# Process Controls of the Live Root Zone and Carbon Sequestration Capacity of the Sundarbans Mangrove Forest, Bangladesh

**Edwin J. Bomer** [1,2,*], **Carol A. Wilson** [1,2] **and Tracy Elsey-Quirk** [2,3]

1   Department of Geology and Geophysics, Louisiana State University, Baton Rouge, LA 70803, USA; carolw@lsu.edu
2   Coastal Studies Institute, Louisiana State University, Baton Rouge, LA 70803, USA; tquirk@lsu.edu
3   Department of Oceanography and Coastal Sciences, Louisiana State University, Baton Rouge, LA 70803, USA
*   Correspondence: ebomer1@lsu.edu

**Abstract:** The conservation of coastal wetland ecosystems, like mangrove forests and salt marshes, represents a critical strategy for mitigating atmospheric emissions and climate change in the 21st century. Yet the existence of these environments is threatened by human-induced disturbances, namely deforestation and accelerated sea-level rise. Coastal systems maintain surface elevation in response to sea-level rise through a combination of physical and biological processes both above and below the ground surface. The quantification and relative contribution of belowground process controls (e.g., seasonal water content, organic matter decomposition) on surface elevation change is largely unexplored but crucial for informing coastal ecosystem sustainability. To address this knowledge deficit, we integrated measurements of surface elevation change of the live root zone (0.5 to 1 m depth) with geotechnical data from co-located sediment cores in the Sundarbans mangrove forest (SMF) of southwest Bangladesh. Core data reveal that the primary belowground controls on surface elevation change include seasonal fluctuations in pore-water content and the relative abundance of fine-grained sediments capable of volumetric expansion and contraction, supporting an elevation gain of ~2.42 ± 0.26 cm year$^{-1}$. In contrast to many mangrove environments, the soils of the SMF contain little organic matter and are dominantly composed (>90%) of inorganic clastic sediments. The mineral-rich soil texture likely leads to less compaction-induced subsidence as compared to organic-rich substrates and facilitates surface equilibrium in response to sea level rise. Despite a relatively high soil bulk density, soil carbon (C) density of the SMF is very low owing to the dearth of preserved organic content. However, rates of C accumulation are balanced out by locally high accretion rates, rendering the SMF a greater sink of terrestrial C than the worldwide mangrove average. The findings of this study demonstrate that C accumulation in the SMF, and possibly other alluvial mangrove forests, is highly dependent on the continued delivery of sediment to the mangrove platform and associated settings.

**Keywords:** climate change; sea-level rise; mangrove soils; surface elevation change; carbon storage

## 1. Introduction

Mangroves forests, and the ecosystems that they foster, are among the most valuable ecological and economic resources on Earth. Mangroves confer a wide range of benefits that improve human livelihoods, including providing a source of food and timber [1], attenuating cyclone-induced storm surges [2,3], and promoting land building through sediment capture [4,5] and below-ground root

growth and leaf litter accumulation [6–8]. Additionally, mangroves and other coastal vegetation represent a significant sink for atmospheric $CO_2$ [9], and they will play a critical role in offsetting accelerated greenhouse gas emissions and resultant sea level rise in the 21st century [10,11]. Despite the general recognition of these benefits, mangroves are in a state of rapid decline: the global areal extent of mangrove forests is decreasing at an average rate of 1% per year [12] and 40% of mangrove tree species are at risk of extinction [13]. While deforestation contributes to the degradation of mangroves, existing research also identifies sea level rise as a primary threat to their survival (e.g., [14,15]), providing numerous examples of mangrove soil surfaces failing to keep pace with the combined effects of sea level rise and subsidence (i.e., "relative sea-level rise" [16–19]). Research efforts that quantify mangrove surface elevation dynamics in response to relative sea-level rise will be paramount for assessing the sustainability of these valuable ecosystems now and into the future.

Among the studies that have investigated the surface elevation dynamics and sustainability in mangrove systems, many have focused on changes in the sediment profile extending from the ground surface to the depth of incompressible substrate (e.g., consolidated sand, limestone, volcanic rock), generally occurring between 5 and 15 m belowground (e.g., [17,18,20,21]). In contrast, comparatively little attention has been given to the uppermost half-meter to meter of the sediment profile, termed the "live root zone" (LRZ), where biological processes have a relatively high influence on surface elevation change [8,22,23]. The LRZ, owing to its relatively shallow position in the subsurface, is particularly sensitive to environmental disturbances that alter surface elevation. For instance, [24] found that mangrove mortality associated with the landfall of Hurricane Mitch caused widespread root decomposition and peat collapse, ultimately resulting in increased shallow subsidence and losses in surface elevation as much as 1.1 cm year$^{-1}$. Even subtle changes in surface elevation can dramatically change the frequency, duration, and depth of inundation by tidal waters [25,26], which can lead to intolerable levels of root submergence and tree death [6].

Within the LRZ is a site-specific mixture of mineral matter, pore-water and gases, and organic matter that includes roots, rhizomes, leaf litter, living organisms, and fine particulate organic material [27]. Compared to terrestrial plants, mangroves are commonly cited to sequester proportionally more carbon belowground (i.e., in the LRZ and below) than aboveground (e.g., [28–30], and worldwide estimates of the mangrove carbon burial often treat the storage ability of different mangrove ecosystems to be equivalent [31,32]. This, however, is an oversimplified approach that discounts the heterogeneity of mangrove ecosystems [33]. While peat and organic-rich substrates are common in mangrove settings and are well represented in the literature [7,8,17,24,34–36], mineral-rich mangrove environments also dominate in many tropical settings [37,38], which may also represent a large carbon sink despite the low concentration of soil organic matter.

Previous research stresses the importance of considering both physical and biological processes to explain changes in surface elevation of the LRZ (e.g., [6]). Yet, there is currently a paucity of studies that link above-ground physical data (e.g., surface elevation change, hydroperiod) to below-ground process controls (seasonal changes in pore-water abundance, oxidation-reduction conditions, etc.) in the LRZ. In this study, we investigate the relative contributions of sedimentary and biotic parameters on controlling surface equilibrium of the LRZ, using the Sundarbans mangrove forest (SMF) of southwest Bangladesh as a study area. Specifically, we compare surface elevation change to: (1) oxidation-reduction potential, (2) pore-water content, and (3) sediment grain size (Figure 1). Another key objective of the study is to evaluate the below-ground carbon sequestration capacity of the SMF and compare our findings with other mangrove forests across the globe. The proceedings of this research provide new information on the bio-physical dynamics of the LRZ and have implications for the sustainability and carbon storage capacity of mangrove ecosystems worldwide.

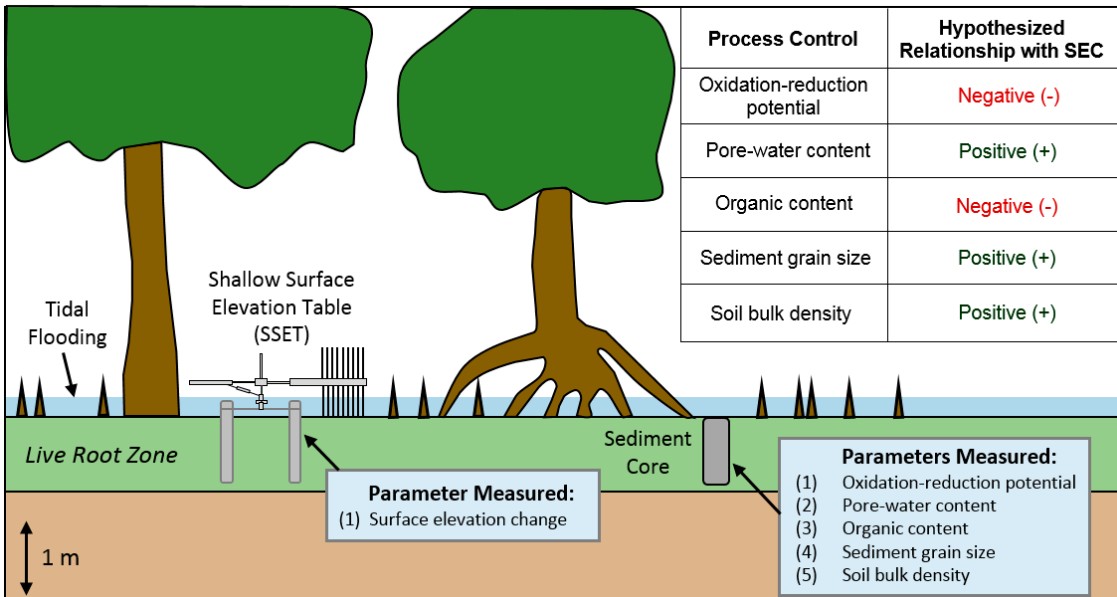

**Figure 1.** Field techniques employed in this study and hypothesized relationships between process controls and surface elevation change (SEC) of the live root zone (LRZ).

## 2. Study Area

The Sundarbans mangrove forest (SMF), encompassing ~10,000 km$^2$ in coastal Bangladesh and India, represents the largest contiguous mangrove forest on Earth (e.g., [39]). The dominant mangrove species and namesake of the forest is the endangered Sundari (*Heritiera fomes*) [40,41]. Other species here include: Bain (*Avicennia marina*), Gewa (*Excoecaria agallocha*), and Kankra (*Bruguiera decandra*). Similar to other mangrove forests, the areal extent of the SMF has changed in response to the level of human activity. In the 18th century, the SMF was roughly twice its present-day size as the northern half of the forest was cleared to expand cultivated land [42,43]. In more recent times, the forest has largely remained intact, exhibiting 1.2% net land loss between 1973 and 2000 [44], and <2% infilled tidal channels since the 1960s [45]. The stability of the SMF can be attributed to preservation measures establishing the forest as a reserve in 1875 (e.g., [43]), as well as the efficient dispersal of fluvial- and marine-sourced sediments by tides throughout the mangrove islands [25,37].

The focus area for this study is located in the northernmost reaches of the SMF, covering ~20 km$^2$ of intertidal mangrove forest, mudflats, and tidal channels (Figure 2). Water and sediment is distributed to the interior of the forest by tidal flooding of the Suterkhali River and its distributive channels (~100–200 m wide, ~5–10 m deep; Figure 2). The forest floor exhibits little topographic change apart from a network of creeks (~1–3 m wide, ~0.2–0.5 m deep) that accommodate the movement of tidal waters. Small-scale surface perturbations on the forest floor include pneumatophore roots, saplings, and mud crab (*Scylla serrata*) mounds. The frequency and duration of platform flooding is seasonally dependent, with higher water levels and ~70% of the annual flooding occurring during the summer monsoon season (June-September; [20]). During the monsoon, water in the local tidal channels is fresh [46]) and exhibits elevated levels of suspended sediment concentration (>1 g/L; [25]). In the dry season (December-April), these tidal channel waters have higher salinity (5–25 ppt; [46]) and lower suspended sediment concentration (0.1–0.6 g/L; [25]). This seasonal pattern also applies to water that floods the forest floor, though suspended sediment concentration is uniformly lower [25], presumably due to particle settling and trapping by vegetation (e.g., [4]). In contrast to the human cultivated landscape to the north, tidal channels in the SMF are not artificially embanked. Hence, this region provides a unique opportunity to investigate surface elevation change and belowground processes of the LRZ under relatively natural conditions.

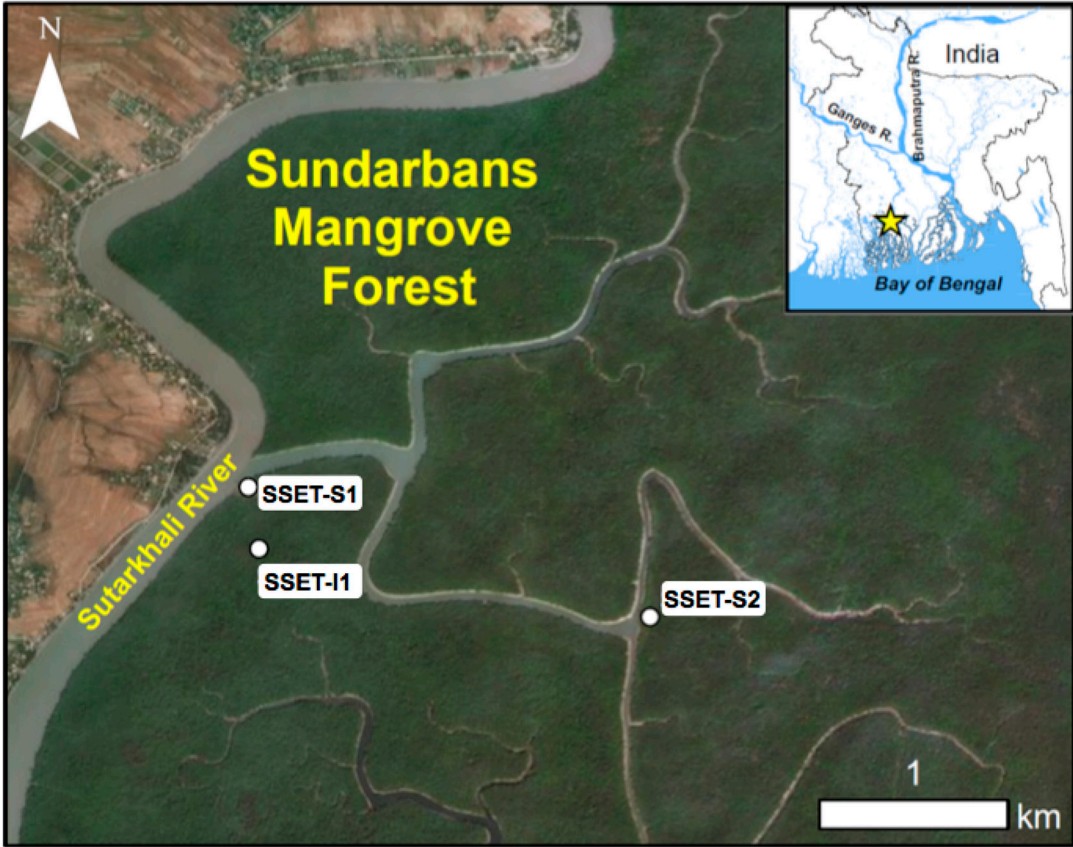

**Figure 2.** Map of the study area with respect to greater Bangladesh and locations of SSET stations. Shallow cores (depth up to 1 m) were taken seasonally within close proximity (<10 m away) of each station.

## 3. Materials and Methods

### 3.1. Site Selection and Near-Surface Elevation Dynamics

Seasonal and inter-annual elevation change of the LRZ was recorded using shallow surface elevation table (SSET) instruments, a variant of the surface elevation table (SET; [47]) composed of four aluminum pipes of 0.75 m length and specifically designed to measure elevation dynamics in the LRZ (Figure 1). SSET's were installed in different hydrodynamic settings to capture the suite of environments comprising the Sundarbans vegetated platform: SSET-S1 is situated adjacent to the Suterkhali River, a first-order tidal channel (stream-bank, higher elevation); SSET-I1 is located in the interior of the mangrove platform, distal (>100 m) from tidal channels (interior, lower elevation); and SSET-S2 is located adjacent to a smaller, second-order tidal channel (interior, higher elevation; Figure 2). Measurements were performed seasonally, immediately after the monsoon season (Oct–Nov) and during the dry season (Mar–May). Mangrove speciation and diversity is consistent among the three sites, though platform hydroperiod data from a recent study help quantify subtle elevation differences that control the amount of tidal flooding and sediment accretion among the locations [20].

### 3.2. Oxidation-Reduction Potential

To assess changes in oxidation-reduction potential (ORP) with depth, cores up to 1 m in length were collected in the vicinity of SSET locations using a 6-cm diameter half-cylinder auger (Figure 1). Large mangrove pneumatophores and belowground roots were avoided during coring to minimize compaction and obtain an undisturbed sediment profile. Cores were returned to the boat where ORP measurements were taken at 10-cm intervals using a handheld gel electrolyte ORP electrode with a

platinum sensing pin (*Hanna Instruments HI3620D*). For calibration purposes, the electrode was soaked in pre-treatment solution for 15 min prior to analysis. ORP measurements were taken by inserting the electrode 2 cm into the sediment interface and by waiting until the reading reached equilibrium. The electrode was rinsed with water following each measurement, and care was taken to shield core sediments from direct sunlight during data collection. Measurements were performed during two monsoon and dry season field campaigns to assess seasonal differences in ORP.

### 3.3. Soil Pore-Water Content, Organic Content, and Bulk Density

Following ORP measurements, 2-cm thick subsamples were taken from cores every 10 cm and packed into air-tight bags. Upon return to the lab, wet sediment samples were weighed and placed in a drying oven at 60 °C for at least 72 h to attain a constant dry mass. The mass of the dehydrated sediment was recorded and compared to the wet weight using Equation (1) to calculate the percent water content, where $m_d$ and $m_w$ refer to the dry and wet sediment mass, respectively.

$$((m_w - m_d)/(m_w)) * 100 \tag{1}$$

Soil total carbon (TC) and total organic carbon (TOC) were quantified by dry combustion in an induction furnace coupled to a Costech 1040 CHN Analyzer (Louisiana State University Wetland Biogeochemistry Laboratory). For TC analyses, 5 to 10 mg of dehydrated and homogenized sediment was sealed in tin capsules and combusted at ~1350 °C for 5 to 7 min [48]. A similar approach was taken for TOC analyses, except sediments were fumigated overnight with HCl vapor before combustion to remove inorganic carbonate minerals (e.g., calcite, dolomite) [49]. Soil organic matter was also assessed by the semi-quantitative loss-on-ignition (LOI) method for comparison [50]. Briefly, dehydrated sediments were homogenized using a mortar and pestle and combusted in a muffle furnace at 550 °C for 5 h to yield weight LOI. Percent organic matter was quantified following Equation (2), where $m_b$ and $m_s$ refer to the biomass and pre-burn sample mass, respectively. For each core, at least one replicate measurement and four QC standards were taken for TC, TOC, and LOI, and samples were re-analyzed if any of the standards deviated more than ±5% from the average.

$$(m_b/m_s) * 100 \tag{2}$$

Dry soil bulk density was calculated by dividing the dry weight of each sediment sample by the volume of the sample (V = 61.33 cm$^3$). Values of soil bulk density were then multiplied by % TOC to obtain soil C density. C sequestration rates were calculated as the product of soil C density and average sediment accretion rates derived from long-term $^{137}$Cs activities (1.1 cm year$^{-1}$; [51]) and short-term sediment tiles (3.0 cm year$^{-1}$; [20]).

### 3.4. Granulometry

Grain size analysis was conducted at 10-cm intervals for all cores (Figure 1). Wet sediment aliquots of ~2 g were placed in test tubes and stirred with 2 mL of 30% hydrogen peroxide ($H_2O_2$) to remove fine organic matter. Following digestion, 15 mL of 0.05% sodium metaphosphate ($NaH_2PO_4$) was added to each solution, stirred to de-flocculate clay particles, and poured through an 850 μm sieve to remove large organic debris (e.g., shells, crab claws). Samples were then ultrasonically dispersed in a Beckman-Coulter laser diffraction particle size analyzer (Model LS 13 320) to calculate the relative abundance of grain sizes between 0.4 and 850 μm. The volumetric abundance of cohesive sediments for a particular sample was taken as the percentage sum of particle sizes <20 μm [52].

### 3.5. Statistical Analyses

Linear regression models were performed on the trends of dependent variables (e.g., pore-water content, surface elevation change, grain size) to ascertain whether the relationships between variables

were statistically significant. Seasonal differences of process controls, including surface elevation change, water content, organic content, and oxidation-reduction potential, were tested for statistical significance using a two-sided t-test assuming equal variances (*sensu* [18]). Following the procedure of [17], a significance level of $\alpha = 0.05$ was compared to the P-value to accept or reject the null hypothesis.

## 4. Results

### 4.1. Near-Surface Elevation Dynamics

Inter-annual trends in surface elevation change of the LRZ demonstrated positive elevation change through time at all SSET locations (Table 1; Figure 3). The annual rate of elevation change was similar at SSET-S1 and SSET-I1 (mean ± standard error = $2.06 \pm 0.17$ cm year$^{-1}$) but higher at SSET-S2 ($3.14 \pm 0.46$ cm year$^{-1}$, Table 1). Two distinct signatures in seasonal elevation change were identified: (1) seasonal step-wise increase or (2) non-seasonal near-linear increase (Figure 3). SSET-S1 and I1 exhibit the former trend and are characterized by greater elevation gain following the summer monsoon season ($2.06 \pm 0.09$ cm) as compared to after the dry season ($0.26 \pm 0.08$ cm) (Table 1; Figure 3). Differences in seasonal elevation change were statistically significant at SSET-S1 ($t$-ratio = 9.01; $p < 0.001$) and SSET-I1 ($t$-ratio = 3.29; $p = 0.022$). Conversely, SSET-S2 does not display any evidence of seasonal variability in elevation change ($t$-ratio = 0.42; $p > 0.05$), exhibiting similar values after the monsoon ($1.70 \pm 0.22$ cm) and dry seasons ($1.44 \pm 0.24$ cm) (Table 1; Figure 3).

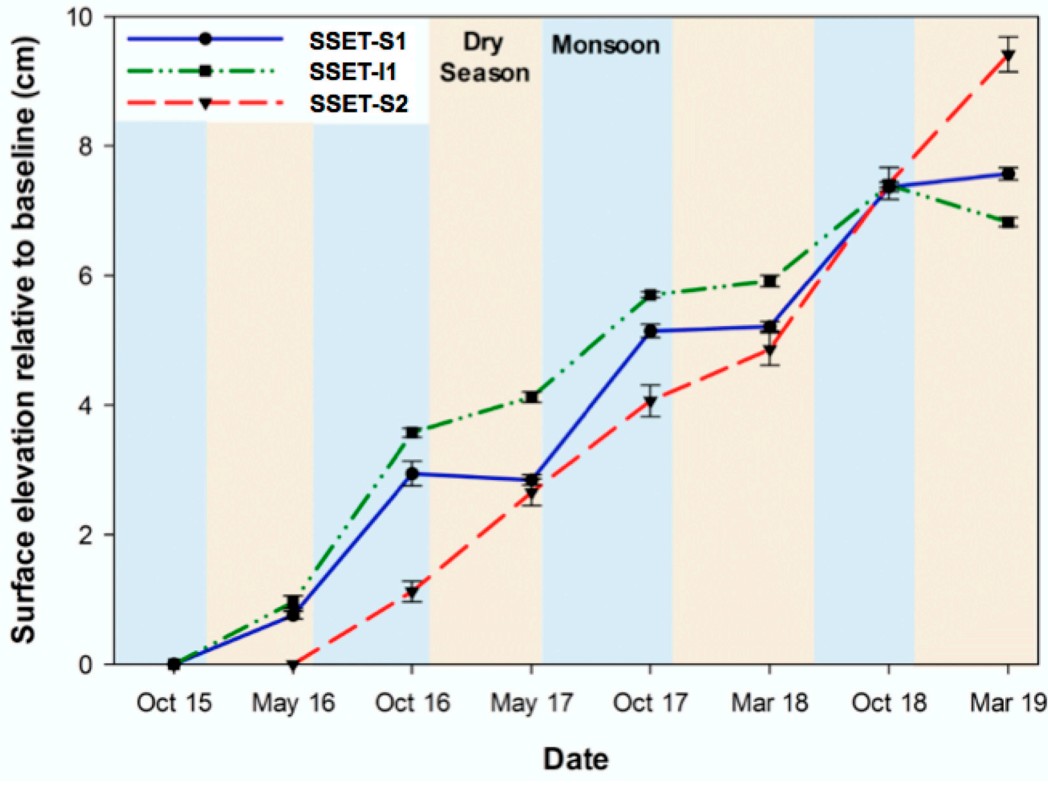

**Figure 3.** Inter-annual change in surface elevation of the live root zone. Values represent longitudinal change relative to the baseline measurement. Error bars are the standard error for all measurements.

**Table 1.** Seasonal and inter-annual rates (± standard error) of surface elevation change (SEC) in the live root zone.

| Station | Record (year) | Mean Annual SEC (cm year⁻¹) | Mean Monsoon SEC (cm year⁻¹) | Mean Dry Season SEC (cm year⁻¹) |
|---|---|---|---|---|
| SSET-S1 | 3.5 | 2.16 ± 0.19 | 2.22 ± 0.15 | 0.23 ± 0.08 |
| SSET-I1 | 3.5 | 1.95 ± 0.14 | 1.89 ± 0.05 | 0.29 ± 0.08 |
| SSET-S2 | 3.0 | 3.14 ± 0.46 | 1.70 ± 0.22 | 1.44 ± 0.24 |

### 4.2. Oxidation-Reduction Potential

ORP measurements from sediment cores varied considerably depending on season, core location, and core depth, showing an overall range of −156 to +158 mV (Figure 4). Cores taken during the monsoon season demonstrated uniformly reduced conditions (i.e., <0 mV) throughout the sediment profile, ranging between −156 and −17 mV among all cores (Figure 4). ORP trends for cores at SSET-S1 and S2 exhibited minimal variance with depth (mean = −140 mV) (Figure 4). SSET-I1, on the other hand, displayed progressive oxidation from the surface (Eh = −156 mV) to 40 cm depth (peaking at Eh = −17 mV) followed by reducing conditions to 90 cm depth (decreasing to Eh = −133 mV) (Figure 4). On the whole, soil conditions were more oxidized in the dry season relative to the monsoon season, ranging between −140 and +158 mV among all cores (Figure 4). The most notable down-core trend in dry season ORP is that the uppermost 30 cm of the soil profile was substantially more oxidized (mean = −4 mV) than depths below 30 cm (mean = −105 mV) (Figure 4). Apart from this trend, ORP measurements below 30 cm were generally invariant with depth in the dry season (Figure 4). Significant seasonal differences in ORP were observed at SSET-S1 (*t*-ratio = 2.52; *p* = 0.0227) and SSET-I1 (*t*-ratio = 2.29; *p* = 0.0341) locations, but not at SSET-S2 (*t*-ratio = 1.45; *p* > 0.05).

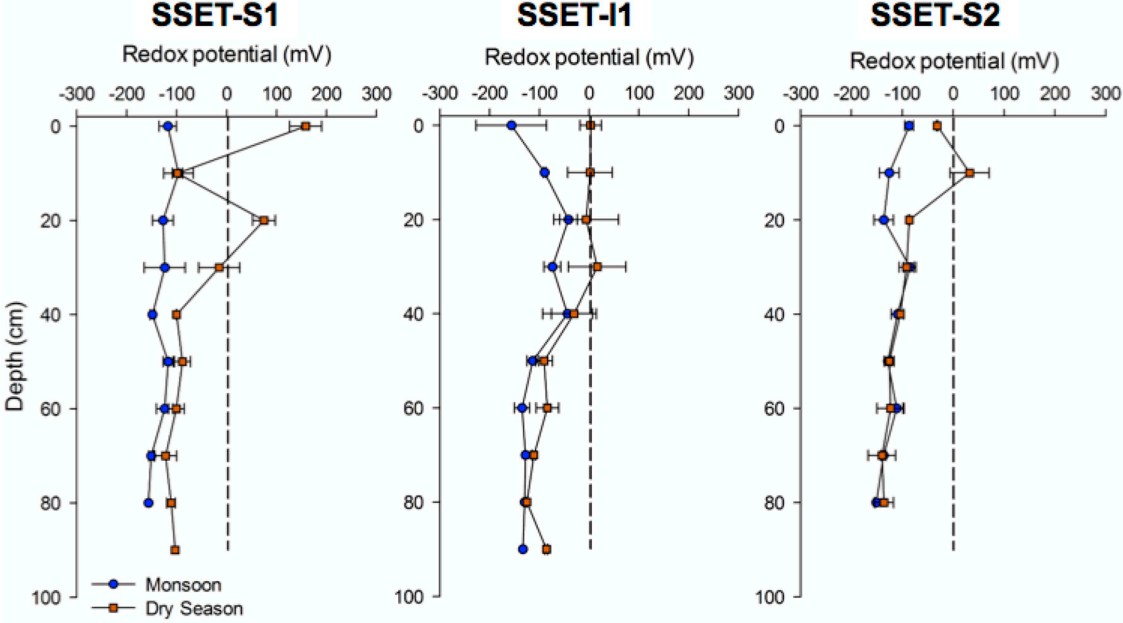

**Figure 4.** Seasonal oxidation-reduction potential of shallow core sediments. Each data point represents the average measurement from replicate cores during four field seasons. Error bars correspond to the standard error of the measurement components. The dashed vertical line indicates the boundary between reduced (negative mV) and oxidized (positive mV) soil conditions. Note that soils are generally reduced with the exception of the upper ~20–30 cm in the dry season.

### 4.3. Soil Pore-Water Content, Organic Content, Bulk Density, and Sequestration Rates

Pore-water content in the LRZ varied primarily based on season, and to a lesser extent, core location and depth (Figure 5). Seasonal differences in pore-water content were found to be significant in cores taken near SSET-S1 ($t$-ratio = 9.39; $p < 0.0001$) and SSET-I1 ($t$-ratio = 11.9; $p < 0.0001$). For these locations, the average pore-water content during the monsoon and dry seasons were 33.4% and 26.1%, respectively. However, seasonal variability in pore-water content is not evident for cores taken near SSET-S2 ($t$-ratio = 1.73; $p > 0.05$). Here, average pore-water content during the monsoon and dry seasons were roughly equivalent at 28.4% and 26.7%, respectively. Trends in down-core water content were not seen in any of the cores apart from a slight increase in pore-water with depth for cores taken near SSET-S2 during the monsoon season (Figure 5).

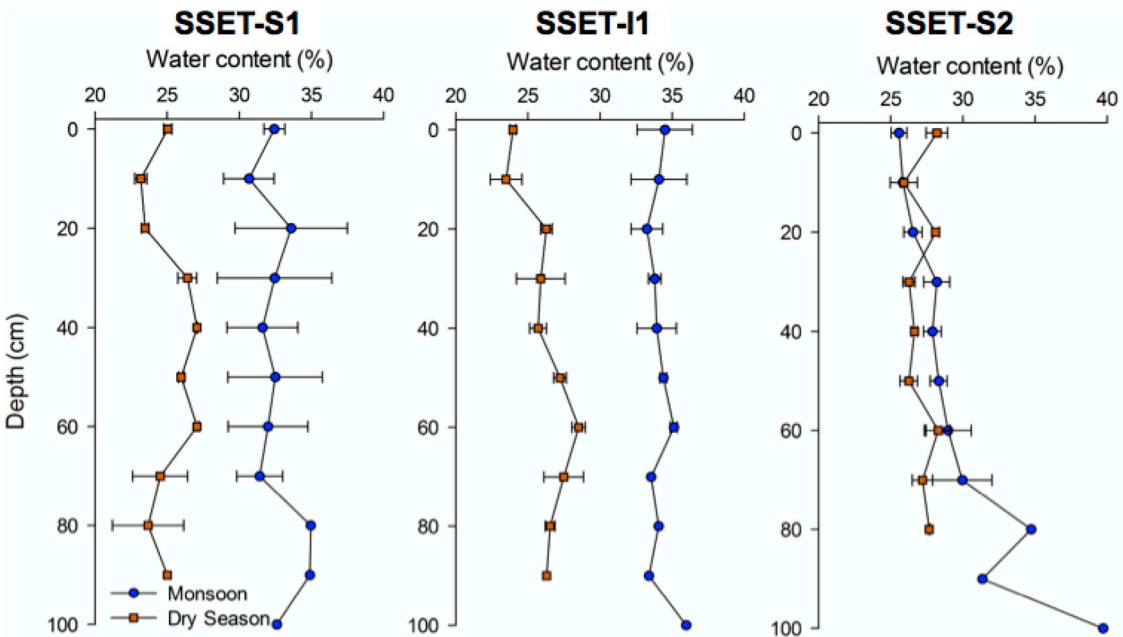

**Figure 5.** Seasonal pore-water content of shallow core sediments. Each data point represents the average measurement from replicate cores during four field seasons. Error bars correspond to the standard error of the measurement components. Note that seasonal differences are statistically significant at SSET-S1 and SSET-I1, but not at SSET-S2.

TC and TOC analyses demonstrate uniformly low organic matter content throughout the shallow subsurface, averaging 1.2 ± 0.1% and 0.9 ± 0.1%, respectively, among all samples (N = 56, Figure 6A,B). No evident down-core trends were observed, and no significant differences in TC or TOC values existed among the coring locations. Average TOC values among all cores were 23.6% lower than average TC values, indicating the presence of inorganic carbon in SMF soils. Organic content as measured by LOI ranged from 3.7% to 6.5% (Figure 6C). Dry bulk density of the shallow subsurface ranged from 0.6 to 1.0 g cm$^{-3}$, with an average value of 0.81 ± 0.08 g cm$^{-3}$. Bulk density did not display any notable trend with depth at any of the core locations (Figure 6D). Core-average soil C sequestration rates calculated using sediment accretion rates from [20] were similar among the three locations, ranging from 200 to 275 g C m$^{-2}$ year$^{-1}$.

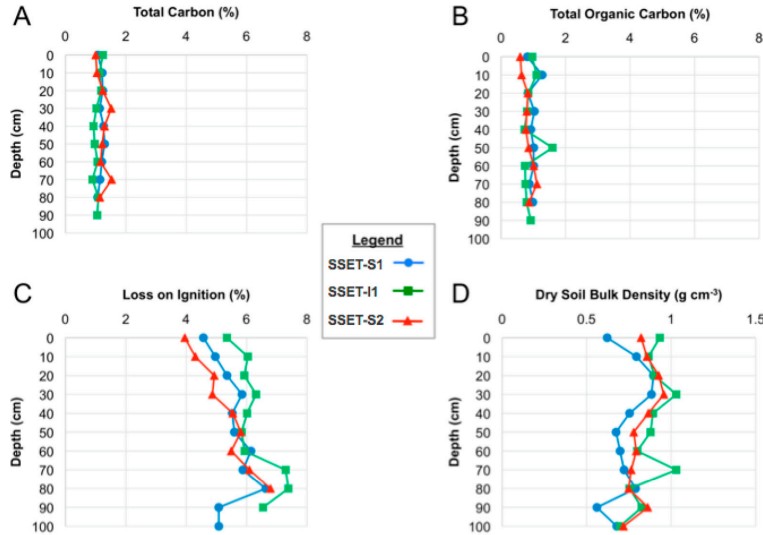

**Figure 6.** Down-core changes in soil (**A**) total carbon, (**B**) total organic carbon, (**C**) loss-on-ignition, and (**D**) dry bulk density.

### 4.4. Granulometry

Grain size analyses of sediments in the LRZ illustrate that median grain size ($D_{50}$) and volumetric abundance of cohesive particles varies depending on core location and depth (Figure 7). Although the median grain size of all core sediments was of medium silt size (16–32 μm), there were differences in average grain size based on core location (Figure 7). Cores at SSET-S2 exhibited the coarsest average grain size (27.7 μm), followed by SSET-S1 (23.9 μm), and SSET-I1 (18.9 μm). Accordingly, cores near SSET-S2 contained the lowest volumetric abundance of cohesive sediment (37.9%), followed by SSET-S1 (45.0%) and SSET-I1 (54.0%) (Figure 7). All cores exhibited a slight coarsening-upward profile, though still fall into the medium silt range (Figure 7).

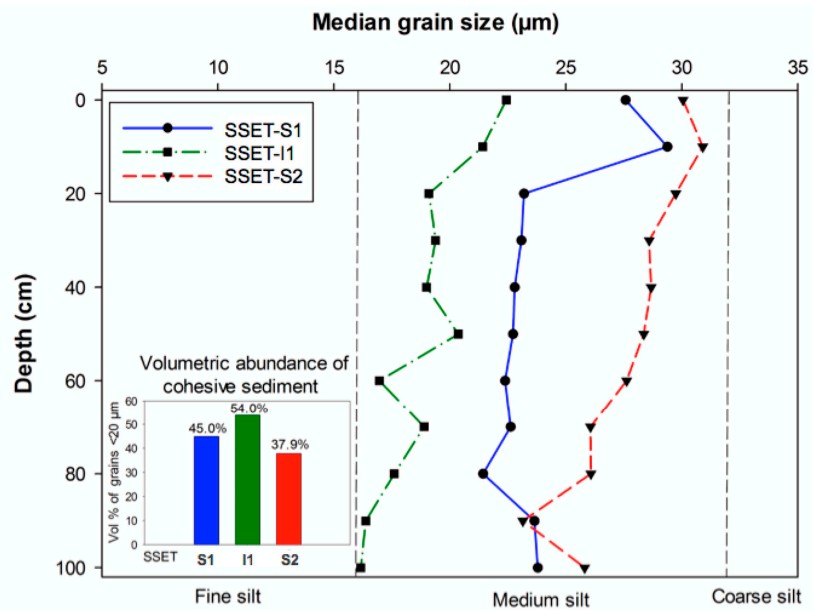

**Figure 7.** Granulometry of sediment cores taken near respective SSET stations. Inset bar graph illustrates the volumetric abundance of cohesive sediments, taken to be particles finer than 20 μm [52]. Note that all cores exhibit a slight coarsening upward succession of median grain size.

## 5. Discussion

### 5.1. Effects of Below-Ground Process Controls on Surface Elevation Change

Studies reporting the relative importance of subsurface processes on changes in surface equilibrium are vital for understanding how mangrove forests and other coastal ecosystems will adjust to environmental disturbances, like locally accelerated sea-level rise [53]. Our results indicate that seasonal changes in soil pore-water storage and oxidation-reduction potential, as well as spatial differences in sediment texture, all influence elevation dynamics in the SMF (Figure 8). For example, seasonal soil pore-water content exhibits a positive and significant relationship with changes in surface elevation of the LRZ ($r^2$ = 0.77, $p$ = 0.0218, Figure 8A). Greater soil pore-water storage during the wet season (Figure 5 or Figure 8A) likely reflects increased platform hydroperiod [20] and infiltration from monsoonal rainfall [54]. A recent study in the SMF found that the mangrove platforms are inundated 46% of the time during the monsoon season, compared to 17% during the dry season [20]. The combination of tidal inundation and rainfall infiltration raises the groundwater table and saturates the substrate, ultimately causing elevation gain via soil swelling (i.e., "dilation water storage," [55,56]). Conversely, decreased soil pore-water content during the dry season (Figure 5 or Figure 8A) is caused by evapotranspiration [57] and lowering of the water table [20], contributing to less elevation gain or even elevation loss (Figure 3, Table 1). These trends did not apply to all locations, however, as SSET-S2 exhibited substantial elevation gain but minimal changes in seasonal pore-water content (Figure 5 or Figure 8A). This may be due to differences in soil texture: SSET-S2 contains a lower abundance of cohesive sediments (Figure 7) and therefore may have better soil drainage in comparison to SSET-S1 and SSET-I1 sites. Seasonal expansion and contraction of the shallow subsurface due to fluctuations in pore-water content and groundwater level have been observed elsewhere, notably in the wetlands of the Florida Everglades [36] and southeast Australia [23,58]. For instance, [58] found in the mangrove forests of Homebush Bay, Australia, that incremental changes in surface elevation over a four-month period were strongly correlated to groundwater depth, which in turn was influenced by the magnitude of monthly precipitation. It follows that locations with seasonal precipitation regimes are more likely to display marked seasonal differences in surface elevation change (e.g., SSET-S1 and I1, Figure 3, Table 1). In southwest Bangladesh, ~80% of the annual rainfall occurs between May and September [54], supporting the idea that seasonal variability in pore-water storage could form a control on local shallow surface elevation change.

Another parameter that appears to influence shallow elevation dynamics is sediment grain size and, more specifically, the abundance of clay minerals and other cohesive sediments capable of volumetric change (e.g., [59]). Although the soil pore matrix hosts a large proportion of groundwater, a significant positive relationship ($r^2$ = 0.62, $p$ < 0.0001) was observed between the abundance of cohesive sediment (<20 μm; [52]) and average pore-water content (Figure 8B), suggesting accommodation of water in the mineral structure of clays and other fine-grained particles (e.g., [60,61]). The potential for soils to exhibit shrink-swell dynamics, and therefore impact shallow elevation change, depends on the local clay assemblage, as certain clay minerals (e.g., smectite and illite) are more prone to hydro-expansion and contraction than others (e.g., kaolinite and chlorite) [27]. Ref. [62] showed that the soils of the SMF are dominantly composed of illite (~60%) with smaller and roughly equivalent proportions of smectite, kaolinite, and chlorite (10–15% each). Given that illite and smectite together account for >70% of the total clay assemblage, the soils of the SMF are likely to exhibit moderate to high shrink-swell characteristics [27]. Areas with a locally high abundance of fine-grained sediments in the LRZ are therefore likely to exhibit seasonal fluctuations in surface elevation and pore-water content, as is observed at SSET-S1 and I1 (Figures 3 and 5). The relatively low abundance of cohesive sediments at SSET-S2 (Figure 7) may explain why this location does not display seasonal differences in surface elevation change (Figure 3; Table 1).

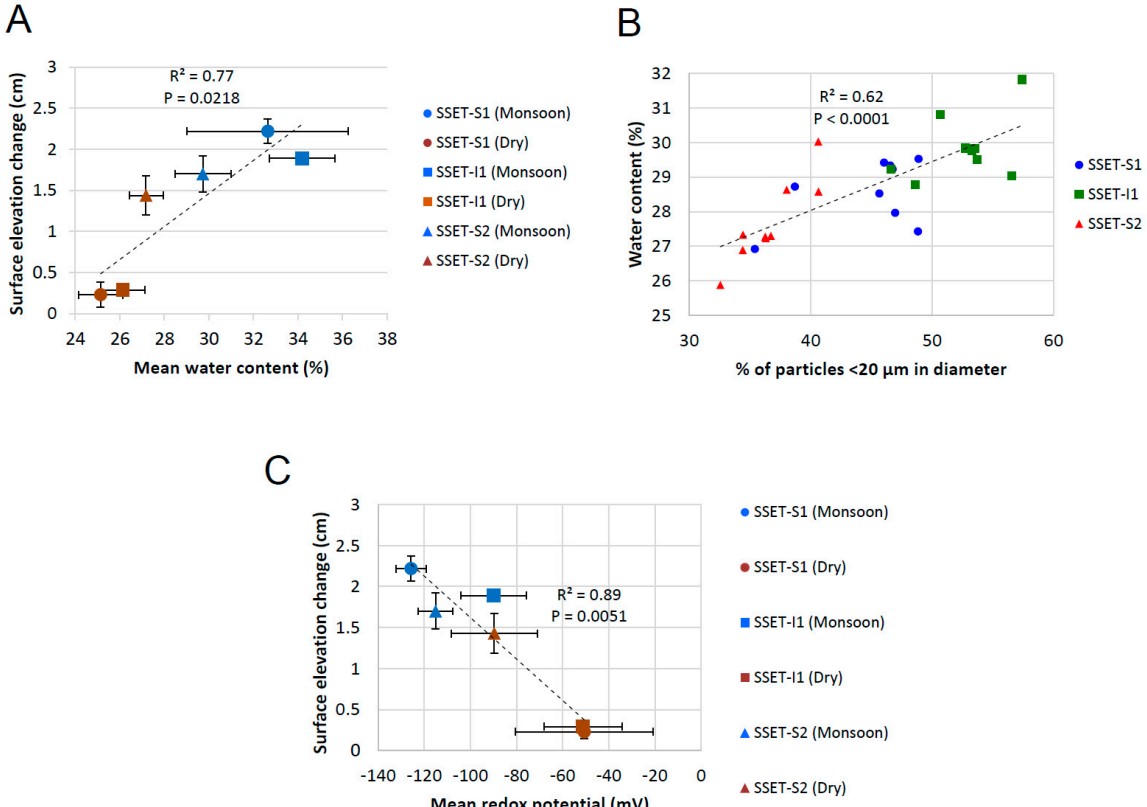

**Figure 8.** Relationships between (**A**) seasonal water content of core sediments and elevation change, (**B**) volumetric abundance of cohesive sediment and water content, and (**C**) seasonal oxidation-reduction potential and elevation change. Particle size data, which does not vary by season, is derived from cores extracted during the monsoon season of 2016. Error bars correspond to the standard error for all associated measurements.

Hydroedaphic conditions, including soil redox potential, have long been recognized as an important control on the physicochemical composition of soils, especially with respect to the preservation or decomposition of organic matter (e.g., [7,63,64]). In general, vascular plant tissue is less likely to be preserved under aerobic soil conditions, as high oxygen levels enhance respiration and microbial activity responsible for the breakdown of lignin and other structural polymers (e.g., [65]). Measurements of soil ORP are thus pertinent for assessing changes in surface elevation as oxidized conditions facilitate the degradation of organic matter (e.g., roots, pneumatophores, buried litterfall) and can lead to compaction-induced shallow subsidence (e.g., [6,24,66,67]). In this study, oxidation of the uppermost ~30 cm of the soil profile was observed at all SSET locations during the dry season (Figure 4) and likely occurs in response to reduced hydroperiods and lowering of the groundwater table [20]. Mean seasonal ORP demonstrates a negative and significant relationship with seasonal shallow elevation change ($r^2 = 0.89$, $p = 0.0051$, Figure 8C), suggesting the potential for enhanced soil aeration to contribute to surface elevation loss via degradation and remineralization of dead biomass. Seasonal changes in ORP vary depending on location: soils at SSET-S1 and SSET-I1 exhibit similar and highly seasonal differences in ORP while those at SSET-S2 demonstrate comparatively little seasonal change (Figures 4 and 8C). Similar to pore-water storage, these site-specific responses may reflect differences in groundwater hydrology that are induced by sediment grain size and texture (Figure 7). Alternatively, a different parameter, like mangrove root production and density, may explain these trends (e.g., [7]).

Dry bulk density values of the shallow subsurface in this study (mean = 0.81 g cm$^{-3}$, Figure 6D) are considerably higher than those reported for other Indo-Pacific mangrove soils

(mean = 0.44 g cm$^{-3}$, [29]), reflecting the preponderance of relatively dense mineral matter and the scarcity of organic matter in the SMF soil profile (Figure 6A–C). High soil bulk density may facilitate the maintenance of surface elevation as clastic sediments are less compressible than organic matter and therefore incur less compaction-induced subsidence with burial [68,69]. Direct observational compaction data is sparse, but it is widely thought that most compaction in wetland soils occurs in the upper 1–2 m of the soil profile (e.g., [70,71]), as measured here with SSET's and core data. With regard to compositional differences, numerical forward models suggest that stratigraphic profiles containing mainly peat compact and subside much more readily than those composed exclusively of silts and sands [72]. Large differences in bulk density within stratigraphic components (e.g., sand overlying peat) were also found to accelerate rates of compaction and subsidence [72]. This does not apply to the SMF system, given its generally homogeneous sediment character (Figure 7), but may be relevant to oceanic mangrove settings, where organic-rich peat is periodically blanketed with offshore sediments delivered by large storm events (e.g., [73]).

While the subsurface parameters investigated in this study (e.g., pore-water content, redox potential, sediment texture) influence shallow surface elevation change (Figure 8), it should be acknowledged that surficial processes (e.g., sediment deposition) could exert a large control on the elevation dynamics of the SMF system. A recent study by [20] quantified the sediment accretion in this region to be 2.97 ± 0.29 cm year$^{-1}$. Comparisons between those data and the rate of shallow surface elevation change determined here (mean = 2.42 ± 0.26 cm year$^{-1}$, Figure 3, Table 1) indicate that ~80% of the shallow elevation dynamics is explained by sediment deposition. Therefore, while pore-water content, redox potential, and sediment texture influence shallow surface elevation change in the SMF system, sediment deposition exerts an overall larger control on the elevation dynamics. This supports that surficial processes generally govern elevation change in sediment-rich, alluvial mangrove environments while subsurface processes appear to play a more prominent role in sediment-deficient, oceanic mangrove systems (e.g., [6,66,67]). Research initiatives aimed at identifying belowground influences on surface elevation change in alluvial mangrove forests are nonetheless important due to site-specific differences that could exist both at regional and global scales.

## 5.2. Carbon Sequestration Capacity of the SMF

The effectiveness of mangrove forests, along with other ecosystems, to function as carbon sinks hinges upon the preservation of organic material during and after burial (e.g., [74]). Total organic carbon (TOC) of SMF soils averaged (± standard deviation) 0.9 ± 0.1% among all samples (Figure 6B), which is considerably lower than soils of other Indo-Pacific alluvial mangrove forests (mean = 7.9 ± 4.6%, [29]). The overall low organic matter content in the LRZ can be explained by a variety of physical and biological mechanisms that operate in the SMF. First, the enormous flux of clastic sediments throughout the Ganges-Brahmaputra (G-B) delta system (e.g., [37,75,76]) likely dilutes the overall abundance of preserved organic matter in the subsurface (e.g., [9,77]). For instance, of the ~1000 Mt of sediment discharged each year by the Ganges and Brahmaputra rivers [78], approximately 100 Mt is deposited in the SMF and G-B tidal delta plain [37]. Consequently, vertical accretion rates in the SMF range between ~2.6 to 3.3 cm year$^{-1}$ [20], ranking among the highest recorded in the literature for mangrove settings [79]. Another possible reason for the mineral-rich soil character is that monsoonal flooding and regular tidal inundation of the forest floor support an energetic hydrodynamic setting capable of entraining and exporting large volumes of organic debris to coastal waters and the open ocean (Figure 9, [80,81]). The extent of tidal flushing evidently extends to the forest interior, as cores taken near SSET-I1 do not contain more organic material than cores taken in proximity to tidal waterways (i.e., SSET-S1 and S2; Figures 2 and 6A–C). This finding emphasizes the efficiency of the SMF drainage network, even within the densely-vegetated interior of the mangrove islands (*sensu* [82]).

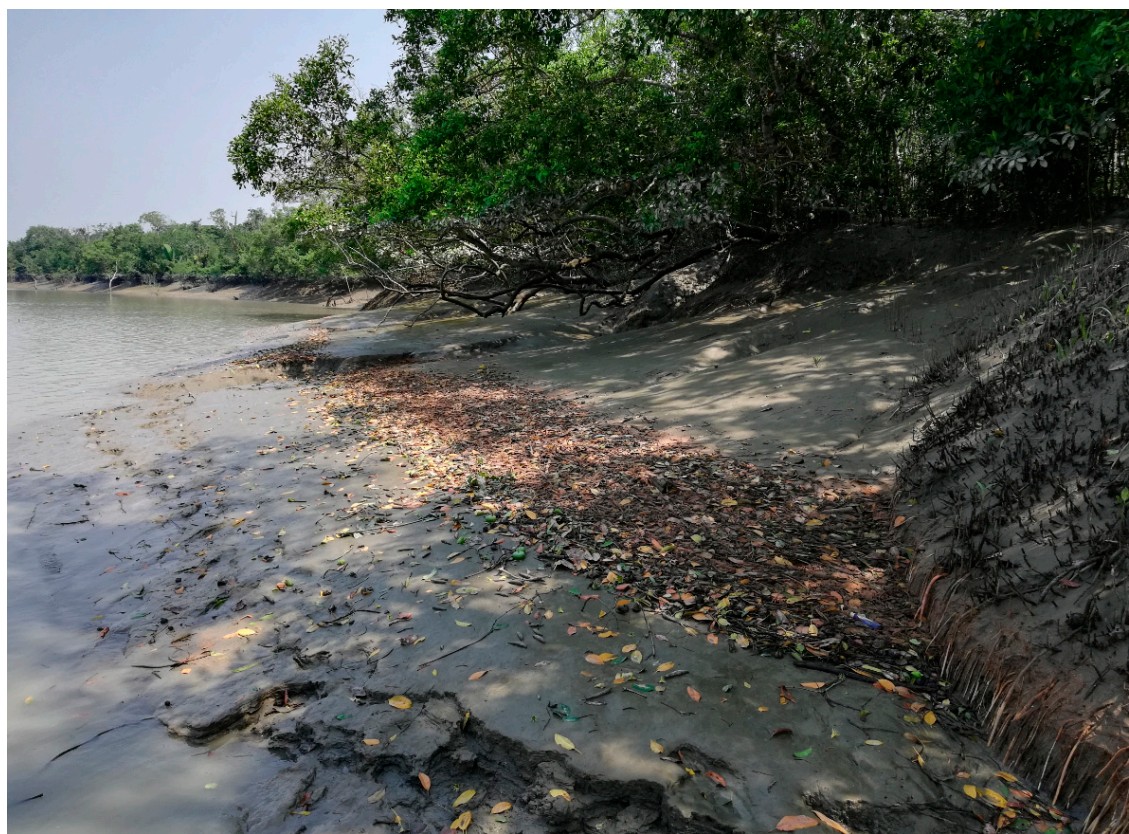

**Figure 9.** Mangrove leaves, twigs, and other organic debris concentrated on the bank of a tidal channel adjacent to site SSET-S2 (see Figure 2 for location). Originally deposited on the intertidal platform, this litterfall is captured by energetic flood tides and is ultimately stored within the tidal channels or is exported to the open ocean.

Organic matter in the LRZ can also be degraded by biological processes, such as leaf-consuming organisms and saprophytic decay (e.g., [64,65,83,84]). While measurements of faunal consumption and organic decay in the SMF are outside the scope of this study (see [85]), the oxidizing conditions observed during the dry season (Figure 4) and general lack of macroparticulate organic matter found in cores supports the concept of dilution, consumption, decomposition, and rapid conversion to small particulate and/or dissolved organic states.

Despite generally high soil bulk density (mean = 0.81 g cm$^{-3}$, Figure 6D), soil C density in the SMF is uniformly low (mean = 0.010 g C cm$^{-3}$, Figure 10A), chiefly due to the scarcity of preserved organic matter in the shallow subsurface (Figure 6A–C). Indeed, the soil C density of the SMF is approximately 82% lower than the worldwide average for mangrove forests (0.055 g C cm$^{-3}$, Figure 10A, [77]). Although spatial heterogeneities throughout the forest are certain to exist, a separate study undertaken in the Indian Sundarbans confirms similarly low soil C densities (0.016 g C cm$^{-3}$, [29]). An assessment of whether the SMF is an effective sink for terrestrial C depends on the rate of sediment accretion and time period considered. For instance, integrating annual-scale sediment accretion rates (~3 cm year$^{-1}$, Figure 10B, [20]) yields C sequestration rates (240 g C m$^{-2}$ year$^{-1}$, Figure 10C) that exceed the worldwide mean for mangrove forests (188 g C m$^{-2}$ year$^{-1}$, Figure 10C, [77], indicating that the locally high rates of sediment accretion compensate for the organic-poor soil quality. This characteristic is mirrored offshore in the Bengal submarine fan, where rapid sedimentation rates and efficient C burial support one of the largest discrete C deposits on Earth–accounting for ~15% of the total terrestrial C buried in oceanic sediments [81]. Application of decadal to centennial rates of sediment accretion as determined by $^{137}$Cs geochronology (1.1 cm year$^{-1}$, [51], however, yields comparatively low C accumulation rates (82 g C m$^{-2}$ year$^{-1}$). As noted in other studies, the period of time over which C

accumulates at the surface and in the subsurface should be carefully considered when evaluating whether ecosystems are productive at sequestering C (e.g., [28,86]). Nonetheless, the calculations of the present study underscore that in the SMF, and possibly other alluvial mangrove forests, continued sediment deposition is critical for the accumulation and sequestration of terrestrial C. Anthropogenic manipulation of sediment-bearing waterways, like the India River Linking Project [87], threatens the current and future potential of alluvial mangrove forests to function as C sinks.

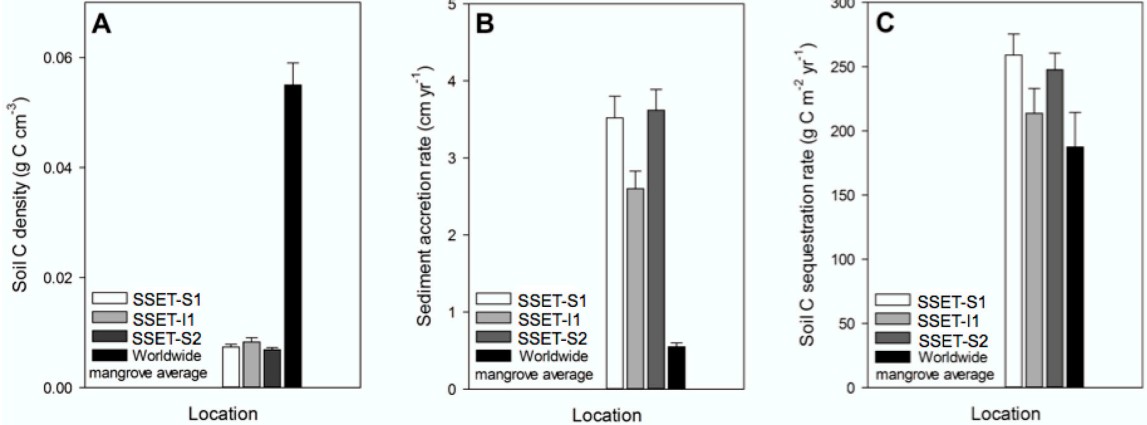

**Figure 10.** Comparisons of mean (**A**) soil C density, (**B**) annual sediment accretion rates, and (**C**) annual soil C sequestration rates among the three SSET locations in this study and the worldwide average for mangrove forests. Error bars represent the standard error for all associated measurements. Soil C density is calculated as the product of soil bulk density and total organic carbon. Rates of C sequestration are calculated as the product of soil C density and the rate of sediment accretion. Data sources: SSET soil C density and soil C sequestration rates (*this study*); SSET sediment accretion rates [20]; worldwide mangrove average soil C density and soil C sequestration rates [77]; worldwide mangrove average sediment accretion rate [79].

There are some limitations to the approach that we took when estimating belowground C densities and sequestration rates, primarily with respect to mangrove root production and biomass. In this study, aerial and belowground mangrove roots were avoided when coring to facilitate retrieval of the full sediment profile. Thus, our measurements of C density and sequestration (Figure 10) only represent the amount of C sequestered by soil and underestimate the total belowground sequestration potential of the SMF. However, it has been shown in a variety of mangrove ecosystems that the majority (~75–90%) of the belowground carbon stock is sequestered within the soil component pool [29,88–90]. Moreover, to ensure consistent analyses, we compared our rates of carbon sequestration exclusively with studies that reported values of organic content from soil ([77] and references therein). Measurements of belowground root and pneumatophore biomass are very difficult to compare owing to a wide variety of sampling procedures. These differences appear to be manifested in data published in the literature. For instance, an extensive review of mangrove biomass studies found a wide range of reported belowground root biomass values (mean ± standard deviation, 78.6 ± 94.6 Mg/ha), relative to soil organic content values (446.9 ± 175.4 Mg/ha) [88]. Therefore, while root production biomass certainly constitutes a portion of belowground carbon storage, the lack of standardization in the associated sampling and analytical methodologies makes comparisons among different ecosystems nebulous. To refine estimations of total belowground biomass and carbon stocks, future studies would benefit from integrating root allometry and soil C measurements.

## 6. Conclusions

Although aboveground depositional processes like sedimentation and organic litter accumulation are critical components of surface elevation change, this study demonstrates that the role of belowground process controls should be taken into account when analyzing and interpreting elevation dynamics. Belowground process controls can be of physical or biological origin and vary in importance depending on the environmental conditions of the system involved [7,33]. In the SMF of southwest Bangladesh, both physical (e.g., seasonal pore-water content and particle size distribution) and biotic parameters (e.g., organic matter decomposition and remineralization) influence subsurface dynamics, despite its status as a mineral-rich deltaic system where physical processes are expected to take precedence. Owing to high inputs of riverine sediments and seasonal aeration/oxidation of the shallow subsurface, the soils of the SMF volumetrically contain very little preserved organic matter. Nevertheless, locally high rates of sediment accretion compensate for the organic-poor soil composition, causing C sequestration rates for the SMF to be higher than those of many mangrove systems worldwide. Continued research, especially in poorly-studied alluvial mangrove forests, is recommended for more accurate inventories of mangrove carbon sequestration and storage. Efforts to safeguard this valuable natural resource, along with other coastal ecosystems, are necessary for any hopes of re-shaping the trajectory of greenhouse gas emissions and minimizing the deleterious effects of climate change.

**Author Contributions:** Conceptualization, E.J.B. and C.A.W.; methodology, C.A.W.; validation, E.J.B., C.A.W. and T.E.-Q.; formal analysis, E.J.B.; resources, C.A.W. and T.E.-Q.; data curation, E.J.B.; writing—original draft preparation, E.J.B.; writing—review and editing, C.A.W. and T.E.-Q.; supervision, C.A.W.; funding acquisition, C.A.W. All authors have read and agreed to the published version of the manuscript.

**Funding:** This study was supported by National Science Foundation Coastal SEES grant #1600258.

**Acknowledgments:** We thank the following individuals for their assistance with field and lab work: Abdullah Al Nahian, Sourov Bijoy Datta, Cameron Gernant, Md. Saddam Hossian, Nithy Khair, Arifur Rahman, and Matthew Winters. Much gratitude is given Md. Nazrul "Bachchu" Islam and Pugmark Tours for gracefully handling field logistics.

**Conflicts of Interest:** The authors declare no conflict of interest. The funders had no role in the design of the study; in the collection, analyses, or interpretation of data; in the writing of the manuscript, or in the decision to publish the results.

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
