# Peer review of "Process Controls of the Live Root Zone and Carbon Sequestration Capacity of the Sundarbans Mangrove Forest, Bangladesh"

_sci, doi:10.3390/sci2030054_

Round 1

Reviewer 1 Report

I think this is an important study with important results. It produces new knowledge on the geo-morphology of the Sundarbans. I appreciate the authors for their efforts. However, I suggest they may think about extending their sampling afforts in their future works. The sampling sites were located in a small area in the northeastern area which is mostly inundated only during the spring tides and mostly supports less salt-toleant mangrove plants. The sedimentation rate and hydrology in the eastern, western and near-sea southern zones are quite different. So looking at these areas in their future studies would help us to have a more robust and general idea on the Sundarbans' hydro-geomorpholy. 

Author Response

I think this is an important study with important results. It produces new knowledge on the geo-morphology of the Sundarbans. I appreciate the authors for their efforts. However, I suggest they may think about extending their sampling efforts in their future works. The sampling sites were located in a small area in the northeastern area which is mostly inundated only during the spring tides and mostly supports less salt-tolerant mangrove plants. The sedimentation rate and hydrology in the eastern, western and near-sea southern zones are quite different. So looking at these areas in their future studies would help us to have a more robust and general idea on the Sundarbans' hydro-geomorphology. We sincerely thank the reviewer for his time and constructive feedback. We are indeed planning future studies in other regions of the Sundarbans (e.g., more seaward locations such as Katka and Hiron Point) to address spatial variability in sedimentation and hydrodynamics across the greater Ganges-Brahmaputra delta plain.

Reviewer 2 Report

This study investigated the surface elevational change of live root zone by using SSET, and the relationship between the surface elevational change and soil characters (e.g., ORP, water content, soil particle size, bulk density) in Sundarbans mangroves. This study can provide useful information, and the scope of this study is well within the scope of Sci. However, because several problems could be found (see below), I would suggest that major revision is required before the paper can be accepted.

Major comments
I do not fully understand the stability of SSET (shallow surface elevation table), and therefore, the data accuracy of surface elevation change. RSET (rod surface elevation table) usually installed into the soil until the basic rock for stability. A benchmark depth ranges from several to nearly 20 m, as the author’s previous paper (Bomer et al. 2020 Catena). But, benchmark depth of SSET was only 0.75 m in this study. Is the stability of SSET broadly accepted? If so, OK, but if not, explanation and evidence for stability of SSET (with the references of previous studies showing the stability of SSET) is needed.

While this study focuses on elevational change in live root zone (top 0.75 m soils), the author’s previous paper (Bomer et al. 2020) focuses on top 15-20 m soils. The stations of SSET-1, -2, and -6 in this study may correspond to RSET-S1, -I1, and -S2 in the previous paper. If the stations of RSET and SSET are located closely to each other, …
・If one station has two station names, it would be better to continuously use only one name to avoid future confusion.
・Additional analysis how the water level data derived from measurement using piezometer on elevational change in live root zone can be conducted. I consider that such data can enrich the paper.

For section 5.1, the paragraph structure should be changed. I consider that one of the most important result of this study is that LRZ showed positive elevational change (mean±SD = 2.42±0.26 cm/yr), and 80% of sediment accretion results from change in LRZ. But, these results were shown in the “end” of section 5.1, while a detailed (bit lengthy) discussion of the mechanism of shrink-swell dynamics based on soil characters (porewater content, soil particle size, ORP and bulk density) was emphasized in section 5.1. Comparison of shallow surface elevational change and total sediment accretion needs to be emphasized, and be stated in the Methods, Result, and early part of 5.1 Discussion section.

Discussion section is bit lengthy. This is because 1) topics with no data often emerged (e.g., leaf-consuming organisms), and 2) several data and results are firstly shown in Discussion section, but not in Result section. For example, comparison of soil C sequestration between the SMF and mangroves worldwide (Fig. 10) is firstly shown in Discussion. However, considering that the “objective of the study is to evaluate the below-ground carbon sequestration capacity of the SMF and compare our findings with other mangrove forests across the globe”, such results can be shown in Result section.

Minor comments
In abstract, there is no detailed data on elevational change. At least mean±SD value (2.42±0.26 cm/yr) is needed

How many rods or pipes per one SSET?

P12L9-10, “leaves should fully decompose after approximately one year.”
Middleton and McKee (2001) studied the early decomposition process of leaves, twigs and roots. Considering the particulate organic matter, leaves may not fully “decompose” after 1yr. Middleton and McKee (2001) used the terms “decay” or “degradation”.

I do not fully understand the data accuracy of soil pore-water content. The authors collected 1-m deep core using a 6-cm diameter half-cylinder auger, and 2-cm thick subsamples were taken from cores every 10 cm. Such sampling procedure may often underestimate the pore-water content, because soil pore-water may often largely leak from the soils within the half-cylinder auger. If so, it would be better to note this phenomenon on methods section, and again, better to consider the use of water-level data derived from measurement using piezometer.

Author Response

This study investigated the surface elevational change of live root zone by using SSET, and the relationship between the surface elevational change and soil characters (e.g., ORP, water content, soil particle size, bulk density) in Sundarbans mangroves. This study can provide useful information, and the scope of this study is well within the scope of Sci. However, because several problems could be found (see below), I would suggest that major revision is required before the paper can be accepted. We thank the reviewer for his detailed review of our manuscript and valuable comments. The manuscript has been revised to address the relevant suggestions - we hope that our explanations and clarifications meet your expectations. We have done our best to indicate where/how those changes are reflected in the new manuscript. I do not fully understand the stability of SSET (shallow surface elevation table), and therefore, the data accuracy of surface elevation change. RSET (rod surface elevation table) usually installed into the soil until the basic rock for stability. A benchmark depth ranges from several to nearly 20 m, as the author’s previous paper (Bomer et al. 2020 Catena). But, benchmark depth of SSET was only 0.75 m in this study. Is the stability of SSET broadly accepted? If so, OK, but if not, explanation and evidence for stability of SSET (with the references of previous studies showing the stability of SSET) is needed. The vertical positioning of the SSET has indeed been shown to be stable over time, as demonstrated by Cahoon et al. (2002), who designed the SSET and compared concurrent RSET and SSET elevation trends over time, finding that: “The survey analysis indicates that the change in height of the shallow benchmark relative to the height of the deep bench mark was within the range of errors associated with the measurement techniques used. Hence we conclude that the shallow benchmark was stable” (pg. 737). Moreover, the focus of this study is not on longitudinal elevation change, which would be impacted by vertical displacement of the benchmark, but rather on seasonal elevation trends, which by virtue of their short time frame, are less likely to be impacted in the case of an unstable benchmark. While this study focuses on elevational change in live root zone (top 0.75 m soils), the author’s previous paper (Bomer et al. 2020) focuses on top 15-20 m soils. The stations of SSET-1, -2, and -6 in this study may correspond to RSET-S1, -I1, and -S2 in the previous paper. If the stations of RSET and SSET are located closely to each other, … - If one station has two station names, it would be better to continuously use only one name to avoid future confusion. We agree with this recommendation and have adjusted the nomenclature to be more consistent with our previous study. - Additional analysis how the water level data derived from measurement using piezometer on elevational change in live root zone can be conducted. I consider that such data can enrich the paper. The topic of how above- and below-ground hydrodynamics impact elevation change was addressed in our previous study (Bomer et al., 2020). The primary reason we chose to focus on pore-water content from sediment cores in this study was because of the highly compatible depths of investigation. For example, the SSET and sediment cores both investigate over depths of ~0.75 m (the live root zone), differing from the piezometers, which investigate over a depth of >2 m. For section 5.1, the paragraph structure should be changed. I consider that one of the most important result of this study is that LRZ showed positive elevational change (mean±SD = 2.42±0.26 cm/yr), and 80% of sediment accretion results from change in LRZ. But, these results were shown in the “end” of section 5.1, while a detailed (bit lengthy) discussion of the mechanism of shrink-swell dynamics based on soil characters (porewater content, soil particle size, ORP and bulk density) was emphasized in section 5.1. Comparison of shallow surface elevational change and total sediment accretion needs to be emphasized, and be stated in the Methods, Result, and early part of 5.1 Discussion section. With all due respect to the Reviewer, we disagree with this suggestion. First, aboveground processes such as seasonal sediment accretion and its impacts on elevation change were established in a previous study (Bomer et al., 2020). Second, the SSET method established by Cahoon et al. (2002) was designed specifically for investigating shallow belowground processes, such as elevation dynamics arising from changes in soil pore-water abundance and seasonal soil oxidation. Therefore the focus of this manuscript is on the belowground processes in this relatively understudied but highly important mangrove forest (as the title and introduction highlight). In the Introduction, we have the following text to highlight these points: “In contrast, comparatively little attention has been given to the uppermost half-meter to meter of the sediment profile, termed the “live root zone” (LRZ), where biological processes have a relatively high influence on surface elevation change [8,22,23]. The LRZ, owing to its relatively shallow position in the subsurface, is particularly sensitive to environmental disturbances that alter surface elevation. For instance, [24] found that mangrove mortality associated with the landfall of Hurricane Mitch caused widespread root decomposition and peat collapse, ultimately resulting in increased shallow subsidence and losses in surface elevation as much as 1.1 cm yr−1. Even subtle changes in surface elevation can dramatically change the frequency, duration, and depth of inundation by tidal waters [25,26], which can lead to intolerable levels of root submergence and tree death [6]… Previous research stresses the importance of considering both physical and biological processes to explain changes in surface elevation of the LRZ (e.g., [6]). Yet, there is currently a paucity of studies that link above-ground physical data (e.g., surface elevation change, hydroperiod) to below-ground process controls (seasonal changes in pore-water abundance, oxidation-reduction conditions, etc.) in the LRZ. In this study, we investigate the relative contributions of sedimentary and biotic parameters on controlling surface equilibrium of the LRZ, using the Sundarbans mangrove forest (SMF) of southwest Bangladesh as a study area. Specifically, we compare surface elevation change to: (1) oxidation-reduction potential, (2) pore-water content, and (3) sediment grain size (Figure 1).” Discussion section is bit lengthy. This is because 1) topics with no data often emerged (e.g., leaf-consuming organisms), and 2) several data and results are firstly shown in Discussion section, but not in Result section. For example, comparison of soil C sequestration between the SMF and mangroves worldwide (Fig. 10) is firstly shown in Discussion. However, considering that the “objective of the study is to evaluate the below-ground carbon sequestration capacity of the SMF and compare our findings with other mangrove forests across the globe”, such results can be shown in Result section. Thanks for these suggestions – we generally agree and have adjusted the results and discussion sections in the following ways. We reduced the amount of interpretative detail on leaf-consuming organisms and saprophytic decay in discussion 5.2. We have also added soil C sequestration rates from this study to the results section 4.3. Note that comparisons of our soil C sequestration rates to the worldwide mangrove average were kept in the discussion section. In the context of the scientific method, we feel that comparisons of our data to the findings of other studies are most appropriate to be included in the discussion section. In abstract, there is no detailed data on elevational change. At least mean±SD value (2.42±0.26 cm/yr) is needed Thank you for this recommendation - we have added this information to the abstract. How many rods or pipes per one SSET? Following the design and procedures of Cahoon et al. (2002), each SSET is composed of four aluminum pipes of ~0.75 m length. We have added this information to methods section 3.1 for clarity. P12L9-10, “leaves should fully decompose after approximately one year.” Middleton and McKee (2001) studied the early decomposition process of leaves, twigs and roots. Considering the particulate organic matter, leaves may not fully “decompose” after 1yr. Middleton and McKee (2001) used the terms “decay” or “degradation”. Thank you for this point of clarification. We have corrected the wording to indicate degradation of organic material and not full decomposition as suggested earlier. I do not fully understand the data accuracy of soil pore-water content. The authors collected 1-m deep core using a 6-cm diameter half-cylinder auger, and 2-cm thick subsamples were taken from cores every 10 cm. Such sampling procedure may often underestimate the pore-water content, because soil pore-water may often largely leak from the soils within the half-cylinder auger. If so, it would be better to note this phenomenon on methods section, and again, better to consider the use of water-level data derived from measurement using piezometer. Thank you for this comment. In our experience collecting cores, the sediments of the Sundarbans readily retain pore water during extraction. This is because the shallow subsurface sediments of the Sundarbans are largely composed of silts and clays (e.g., Allison et al., 2003; Hale et al., 2019; Figure 7 of this study) that have relatively low permeability and do not transmit water easily. Thus, we contend that the pore-water data presented in this study is indeed representative of the true soil character. References cited Allison, M.A.; Khan, S.R.; Goodbred, S.L.; Kuehl, S.A. Stratigraphic evolution of the late Holocene Ganges–Brahmaputra lower delta plain. Sediment. Geol. 2003, 155, 317–342, doi:10.1016/S0037-0738(02)00185-9. Bomer, E.J.; Wilson, C.A.; Hale, R.P.; Hossain, A.N.M.; Rahman, F.M.A. Surface elevation and sedimentation dynamics in the Ganges-Brahmaputra tidal delta plain, Bangladesh: Evidence for mangrove adaptation to human-induced tidal amplification. Catena 2020, 187, 104312, doi:10.1016/j.catena.2019.104312. Cahoon, D.R.; Lynch, J.C.; Perez, B.C.; Segura, B.; Holland, R.D.; Stelly, C.; Stephenson, G.; Hensel, P. High-Precision Measurements of Wetland Sediment Elevation: II. The Rod Surface Elevation Table. J. Sediment. Res. 2002, 72, 734–739, doi:10.1306/020702720734. Hale, R.P.; Wilson, C.A.; Bomer, E.J. Seasonal Variability of Forces Controlling Sedimentation in the Sundarbans National Forest, Bangladesh. Front. Earth Sci. 2019, 7, 211, doi:10.3389/feart.2019.00211.

Round 2

Reviewer 2 Report

Well revised, accepted. But, the quality of Fig. 8 is low, because of low-resolution and indistinguishable symbols for Fig 8 A and C.

Author Response

Well revised, accepted. But, the quality of Fig. 8 is low, because of low-resolution and indistinguishable symbols for Fig 8 A and C. Thanks for this recommendation. We have have increased the symbol size for Fig 8 A and C and increased the resolution of the graphic.